# Genomic and Transcriptomic Dissection of Growth Characteristics and Exopolysaccharide-Related Bioactivities in *Lactiplantibacillus plantarum* NMGL2

**DOI:** 10.3390/foods14203520

**Published:** 2025-10-16

**Authors:** Yanfang Wang, Xinyu Bao, Zhennai Yang, Dong Han

**Affiliations:** 1Beijing Advanced Innovation Center for Food Nutrition and Human Health, Beijing Engineering and Technology Research Center of Food Additives, School of Food and Health, Beijing Technology and Business University, Beijing 100048, China; 2Key Laboratory of Food Bioengineering (China National Light Industry), College of Food Science and Nutritional Engineering, China Agricultural University, Beijing 100083, China

**Keywords:** lactic acid bacteria, whole-genome sequencing, RNA-seq, metabolic regulation, exopolysaccharides

## Abstract

Analyzing the biochemical and physiological activities of food microbes using molecular and bioinformatics tools is important, offering profound insights into their safety, functional, and applicational roles in food. In this study, *Lactiplantibacillus plantarum* NMGL2, a well-documented beneficial lactic acid bacteria (LAB) strain, was investigated for its genomic, metabolic, and transcriptomic characteristics. Whole-genome sequencing revealed that this strain possesses a chromosome and two plasmids, with 3320 annotated genes, showcasing pathways involved in carbohydrate metabolism, stress adaptation, and bioactive compound synthesis. Growth studies under various nutritional conditions, including fructose, lactose, exogenous exopolysaccharide (EPS), and soy peptone, demonstrated that nitrogen source alteration significantly enhanced bacterial growth and EPS production. Transcriptomic analysis showed the addition of EPS and soy peptone resulted in similar regulatory patterns, suggesting shared modulation of metabolic pathways, although distinct gene regulation patterns were involved. In contrast, fructose and lactose primarily regulated carbohydrate metabolism without increasing EPS yield. Prophage gene clusters were consistently down-regulated across all experimental conditions, reflecting the strain’s adaptive response. These findings highlight *L. plantarum* NMGL2’s ability to dynamically adjust its metabolism and gene expression in response to environmental and nutritional changes, offering valuable insights for its application in functional foods and probiotics. These results also imply the potential of LAB strains in bioactive compound production and health-related applications through metabolic engineering.

## 1. Introduction

Lactic acid bacteria (LABs) are a diverse group of microorganisms found in a variety of natural environments. LABs have long been utilized in food production, particularly in the fermentation of dairy products, vegetables, and beverages [1]. In addition to their role in food preservation, LABs are valued for their probiotic properties, including modulation of gut microbiota and production of beneficial metabolites [2,3]. Among them, *Lactiplantibacillus plantarum* is widely studied for its adaptability and health-promoting effects [4,5]. Understanding the genomic and functional features of *L. plantarum* can facilitate unlocking the potential in both food and health applications.

*L. plantarum* NMGL2, isolated from traditional fermented dairy, shows promising bioactivities and food applications. It produces antimicrobial peptides (AMPs) with strong antibacterial effects [6], is resistant to cold and acid stresses [7,8]. Additionally, its exopolysaccharides (EPS) modulate gut microbiota, promoting beneficial bacteria and offering protection against inflammatory bowel disease (IBD) by regulating oxidative stress and inflammation [6,9]. These properties position *L. plantarum* NMGL2 as a promising probiotic candidate for functional food development. High-throughput sequencing technologies, including whole-genome sequencing (WGS) and RNA sequencing (RNA-seq), offer invaluable tools for elucidating the biological information of organisms and enable the examination of gene expression profiles under various growth conditions or environmental stresses [10]. While some genes have been studied, the overall functional genomics of *L. plantarum* NMGL2 remains an underexplored area. This gap in knowledge presents an exciting opportunity to discover novel genetic information, regulatory networks, and regulation mechanisms that may govern important traits such as nutrient utilization, bioactive compound production, and other phycological properties.

In this study, we aim to uncover previously unknown genetic elements that contribute to the functionality and application potential of *L. plantarum* NMGL2 using a combination of analytical tools and pipelines. The *L. plantarum* NMGL2 is cultured with various nutritional conditions, which include the use of different carbon sources, such as fructose and lactose, along with an exogenous supply of EPS and soy peptone as the nitrogen source. Through culture studies, whole genome sequencing, and transcriptomic analyses, we aim to investigate the metabolic characteristics of *L. plantarum* NMGL2 and identify the genes involved in the production of bioactive compounds, thereby supporting its potential applications in fermented foods and probiotic utilizations.

## 2. Material and Methods

### 2.1. Strain and Culture Preparation

The strain *L. plantarum* NMGL2 was isolated from traditional Chinese fermented dairy products, sourced from Inner Mongolia, China [7]. This strain was preserved in de Man, Rogosa, and Sharpe (MRS) broth (Becton Dickinson, Sparks, MD, USA) supplemented with 20% (*v*/*v*) glycerol (Shanghai Yuanye Biotech Co., Ltd., Shanghai, China) at −80 °C for long-term storage. Prior to experimental use, the strain was revived by two successive MRS broth incubation for 18 h at 37 °C to ensure optimal growth and metabolic activity.

### 2.2. DNA Extraction and Whole-Genome Sequencing

The extraction and purification of total genomic DNA from overnight cultured bacterial pellets was performed using the TaKaRa MiniBEST Bacterial Genomic DNA Extraction Kit (Takara Bio Inc., Tokyo, Japan) following the manufacturer’s protocol. The concentration and purity of the extracted DNA were assessed using NanoDrop 2000 spectrophotometer (Thermo Fisher Scientific, Wilmington, DE, USA).

Purified genomic DNA was fragmented using S220 ultrasonicator (Covaris, Woburn, MA, USA) to generate appropriately sized fragments for sequencing library construction. The resulting DNA samples were then subjected to library preparation and sequencing on the NovaSeq X Plus platform (Illumina Inc., San Diego, CA, USA) to generate paired-end sequences (2 × 150 bp) results according to manufacturer’s protocol. The raw sequencing data quality was evaluated and filtered with FastQC (Version 0.12.0) [11]. The clean reads were assembled de novo using SOAPdenovo (Version 2.04) to generate the draft genome [12]. The resulting genome assembly was then submitted to the national center for biotechnology information (NCBI) database and subjected to the NCBI prokaryotic genome annotation pipeline (PGAP) for annotation and public documentation [13,14]. After the processing by PGAP, the comprehensive genomic data were deposited into the Proksee web server with the GenBank accession number for circular genome analysis and visualization [15]. Gene function annotations based on Kyoto Encyclopedia of Genes and Genomes (KEGG) Orthology (KO) were carried out using KEGG BlastKOALA (Orthology And Links Annotation) (accessible at http://www.kegg.jp/blastkoala, accessed on 1 September 2025) and pathway reconstructions were conducted using the KEGG database [16,17].

### 2.3. Extraction and Purification of EPS

The EPS was extracted, isolated, and purified by boiling fermented *L. plantarum* NMGL2 MRS broth at 100 °C for 10 min to deactivate any enzymes that might degrade the polymer. An 80% trichloroacetic acid (Shanghai Yuanye Biotech Co., Ltd., Shanghai, China) solution was added to the broth to achieve a final concentration of 4%, and the mixture was stirred at room temperature for 2 h. The cells and precipitated proteins were removed by centrifugation at 9000× *g* and 4 °C for 20 min. The supernatant was then mixed with two volumes of anhydrous ethanol (Beijing Lanyi Chemical Co., Ltd., Beijing, China) and stored at 4 °C for 24 h to precipitate the polysaccharides. The polysaccharide precipitate was collected by centrifugation (9000× *g*, 4 °C, 20 min), and the resulting pellet was dissolved in a small amount of distilled water. The solution was placed in a dialysis bag (molecular weight cutoff of 8000–14,000 Da) (Shanghai Yuanye Biotech Co., Ltd., Shanghai, China) and dialyzed at 4 °C for 48 h to produce crude EPS, with the water changed every 8 h, until no monosaccharides were detected in the dialysis solution by the phenol-sulfuric acid method. The crude EPS was then freeze-dried for 2 days in SCIENTZ-10N/A vacuum freeze-dryer (Scientz Biotechnology Co., Ltd., Ningbo, China). Overnight cultures grown in regular MRS broth were subjected to this extraction and purification procedure to obtain freeze-dried crude EPS derived from *L. plantarum* NMGL2, which were subsequently used as exogenous EPS components in growth studies.

### 2.4. Growth Study Under Different Culture Conditions

Five different culture conditions were employed in incubation studies to investigate growth dynamics as well as transcriptomic regulation changes (Table 1). Detailed formulation and denotations are as follows: Control (also denoted as “C”), regular MRS cultured with glucose as carbon resource. Fructose (also denoted as “F”), MRS cultured modified with fructose as carbon resource. Lactose (also denoted as “L”), cultured with MRS modified with lactose as carbon resource. EPS (also denoted as “E”): regular MRS cultured with glucose and supplementation of purified EPS produced by method described in Section 2.3. Protein (also denoted as “P”): cultured with MRS medium modified by substituting the tryptic digest of beef and bovine bone peptone with papain digested soy peptone.

Revived *L. plantarum* NMGL2 culture was washed twice and resuspended with sterile 0.85% NaCl (Beijing Lanyi Chemical Co., Ltd., Beijing, China). The optical density measured at 600 nm (OD_600_) of the bacterial suspension was measured using DiluPhotometer™ spectrophotometer (Implen, Westlake Village, CA, USA) and adjusted to 1.0 by adding sterile 0.85% NaCl solution as necessary. Growth experiments were initiated by transferring 300 µL of the suspension (OD_600_ = 1.0) into 10 mL growth culture of the 5 groups and incubated at 37 °C for up to 24 h.

### 2.5. Bacterial Cell Density, pH, and Exopolysaccharide Yield Monitoring

Samples from four groups without EPS supplementation were collected at 0, 5, 6, 10, 12, 16, and 24 h to assess bacterial cell density (OD_600_) and pH. Specifically, OD_600_ were measured using the spectrophotometer method described in Section 2.4. The pH of samples was measured using PB-10 pH meter (Sartorius, Göttingen, Germany). For EPS yield measurements, samples collected at 10 h and 24 h were evaluated as follows, the total carbohydrate content of crude EPS extracted from 50 mL of bacterial culture using method described in Section 2.3 was determined using the phenol-sulfuric acid method, with glucose as standard. Briefly, 200 μL of the EPS dialysis solution was mixed with 100 μL of freshly prepared 6% phenol solution (Beijing Lanyi Chemical Co., Ltd., Beijing, China) and 500 μL of 98% concentrated sulfuric acid (Beijing Lanyi Chemical Co., Ltd., Beijing, China). The mixture was then checked for color development and incubated at room temperature for 20 min. After incubation, the sample was thoroughly mixed, and the absorbance was measured at wavelength of 490 nm. The obtained absorbance values were employed to calculate the total EPS yield by referencing the glucose (Shanghai Yuanye Biotech Co., Ltd., Shanghai, China) standards.

### 2.6. Bacterial Population Measured by Plate Counting

After 5 and 10 h of incubation, 100 μL of homogenized cultures from all five groups were subjected to serial decimal dilutions in sterilized peptone saline solution (0.1% peptone, 0.85% NaCl). Population enumeration was performed by plating 100 μL of each dilution onto MRS agar (Becton Dickinson, Sparks, MD, USA) in duplicate, followed by aerobic incubation at 37 °C for 24 h.

### 2.7. RNA Extraction and RNA-Seq

After 10 h of incubation, bacterial transcriptional activities of all 5 groups were terminated by adding an equal volume of pre-chilled isopropanol (Beijing Lanyi Chemical Co., Ltd., Beijing, China). RNA extraction, rRNA depletion, and genomic DNA removal were conducted using the RiboPure™ RNA Purification Kit for Bacteria (Ambion, Austin, TX, USA), the Ribo-Zero™ rRNA Removal Kit (Bacteria) (Ambion, Austin, TX, USA), and a DNase I digestion kit (Takara Bio Inc., Tokyo, Japan), respectively. The reverse transcription and complementary DNA (cDNA) synthesis was carried out using SuperScript™ IV reverse transcriptase (Invitrogen, Carlsbad, CA, USA). The cDNA samples were then subjected to library preparation and sequencing on the NovaSeq X Plus platform (Illumina Inc., San Diego, CA, USA) to generate paired-end sequences (2 × 150 bp) results according to manufacturer’s protocol.

Transcriptome analysis was carried out on the Galaxy platform (accessible at http://usegalaxy.org, accessed on 1 September 2025) [18]. Initial quality control of the raw RNA-seq data was performed using FastQC (Version 0.12.0) for quality checks [11]. Sequence trimming and filtering were carried out with Trimmomatic [19]. RNA-seq reads were aligned to the genome obtained from whole-genome sequencing with PGAP annotation using Bowtie2 (Version 2.5.4) [20]. For downstream analysis, HTSeq and the htseq-count script was employed to calculate gene expression levels of genes with the identifier of CDS (coding sequence) [21,22], while DESeq2 was used for normalized differential gene transcription analysis, distance matrix calculation, principal component analysis (PCA) plotting, and sample-to-sample distance heatmap generation [23]. Circular genome transcription visualization was carried out with ShinyCircos-V2.0 (accessible at https://venyao.xyz/shinycircos, accessed on 1 September 2025) [24,25].

### 2.8. Statistical Analysis

All experiments were conducted with at least three biological replicates. Statistical analyses were performed using SPSS software (Version 21.0.0, SPSS Inc., Chicago, IL, USA). The microbial population counts, determined via plate counting, were log-transformed before analysis. Single-factor analysis of variance (ANOVA) was used to evaluate group differences, and Tukey’s test was applied to assess statistical significance (*p* < 0.05). For RNA-seq, differential expression analysis was conducted using DESeq2 (Version 1.40.2) with Benjamini–Hochberg correction for multiple testing (adjusted *p* < 0.05 considered significant).

## 3. Result and Analysis

### 3.1. Whole Genome Sequencing and Annotation of Lactiplantibacillus Plantarum NMGL2

The WGS of *L. plantarum* NMGL2 was carried out using high throughput sequencing to provide a comprehensive genomic overview. The whole genome of *L. plantarum* NMGL2 consists of a single chromosome (CP172591.1) with a length of 3,323,395 bp and a GC content of 44.5% (Figure 1A). In addition to the chromosome, two plasmids were identified: Plasmid 1 (CP172592.1), 70,687 bp in length with a GC content of 38.5% (Figure 1B); and Plasmid 2 (CP172593.1), 64,268 bp in length with a GC content of 39.5% (Figure 1C).

Based on the PGAP annotation, a total of 3320 genes were identified in the genome, including both chromosomal and plasmid-associated genes. Among these, 3171 genes were predicted as potential protein-coding genes, constituting approximately 95.5% of the total predicted genes. Additionally, 100 RNA genes were identified, including five complete sets of rRNA operons (5S, 16S, and 23S). The chromosome harbors 3185 genes, while Plasmid 1 and Plasmid 2 contain 68 and 67 genes, respectively. Notably, all plasmid-associated genes were predicted to encode proteins, whereas the other types of genes exclusively locate on the chromosome. Furthermore, one CRISPR array was identified on the chromosome (Figure 1).

For functional annotation and analysis, the NCBI basic local alignment search tool (BLAST)-based KEGG annotation was used to examine the 3115 protein-coding genes assigned non-redundant protein accession numbers (WP_). Among these, 1572 genes (50.5%) were mapped to KEGG orthology (KO) groups. The annotated genes were primarily associated with key functional categories, including carbohydrate metabolism, genetic information processing, and signaling and cellular processes (Appendix A).

### 3.2. Culture Dynamics and Exopolysaccharide Production

In the culture dynamics study, *L. plantarum* NMGL2 was inoculated at an identical initial population density into four differently formulated MRS-based culture groups: Control, Fructose, Lactose, and Protein (Figure 2). Key parameters, including OD_600_, pH, and EPS yield, were monitored over a 24 h incubation period. Across all four groups, an increase in OD_600_ was observed after 5 h of inoculation, followed by a rapid growth phase as the bacteria entered the exponential growth period. Significant differences in growth performance were noted after 24 h of fermentation. Specifically, the Fructose group exhibited a significantly lower OD_600_ value (1.42 ± 0.01) compared to the other groups (*p* < 0.05), whereas the Protein group showed the highest OD_600_ value (1.63 ± 0.07), significantly exceeding the other three groups (*p* < 0.05) (Figure 2E). Also, pH values decreased consistently over the fermentation period, reflecting acid production during bacterial growth. After 24 h, significant differences in pH were observed among the groups. The Control group (3.97 ± 0.01) and the Lactose group (3.94 ± 0.01) exhibited the lowest pH values, while the Fructose group displayed the highest pH (4.12 ± 0.01), all significantly different (*p* < 0.05) (Figure 2F). After 24 h of incubation, the Fructose group produced the lowest volume of EPS (147.04 ± 8.61 mg/L), which was significantly lower than the other three groups (*p* < 0.05). In contrast, the Protein group had the highest EPS yield (251.38 ± 8.43 mg/L), significantly exceeding all other groups (*p* < 0.05) (Figure 2G). These trends closely mirrored those observed for OD_600_.

In a parallel transcriptomic study designed to investigate the regulatory effects of EPS accumulation on strain transcription, the same fermentation setups and conditions were employed. Additionally, Control group supplemented with pre-extracted crude EPS from *L. plantarum* NMGL2 (EPS group) was included to simulate conditions of high EPS accumulation. Viable cell counts were analyzed at 5 and 10 h (Figure 2H). Notably, both the Protein group and the EPS group displayed significantly higher viable cell counts at both time points compared to the other groups (*p* < 0.05) except for the Fructose group at 5 h. This trend was particularly pronounced at the 10 h mark, where these two groups demonstrated significantly higher viable counts than the other three groups (*p* < 0.05).

### 3.3. RNA-Seq and Transcriptomic Profiling

A comprehensive transcriptomic analysis of *L. plantarum* NMGL2 was performed using RNA-seq, with the genome assembly and annotation from whole-genome sequencing serving as the reference for the analysis. The summary of the htseq-count results showed that the majority of the input sequences were successfully aligned to the genome with Bowtie2 pipeline. Specifically, only 0.75% ± 0.13% of the input sequences across all 15 samples failed to align to (indicated as “not aligned” sequence numbers) the genome, demonstrating a high level of alignment accuracy (Appendix A). PCA plot (Figure 3A) revealed that the first two principal components (PC1 and PC2) accounted for approximately 75% of the total variance. The PCA plot showcased the distinct separation among the five experimental groups, with tight clustering within the three replicates of each group. This indicates that the culture conditions induced unique transcriptional profiles for each group. The strongest group-to-group similarities were observed between the EPS and Protein groups, as evidenced by both the PCA (Figure 3A) and heatmap analyses (Figure 3B). These results demonstrate that the experimental setup and RNA-seq analytical pipeline effectively captured and delineated the transcriptional profiles of *L. plantarum* NMGL2.

Among the 3220 gene loci annotated as CDS, 3112 loci, accounting for 96.65% of the total CDS, have aligned sequences and are identified as normalized counts in at least 10 samples (Appendix A). This observation suggests that these genes are actively transcribed into mRNA under respective culture conditions. Subsequently, gene regulation comparison was performed on these genes. In the comparisons between individual groups (Figure 4), genes showing a Log_2_ fold change greater than 2 and a -Log_10_ *p*-value exceeding 5 were considered significantly regulated. The results are as follows: 44 genes were up-regulated and 84 genes were down-regulated in the Fructose group compared with the Control group (Figure 4A); 32 genes were up-regulated and 24 genes were down-regulated in the Lactose group compared with the Control group (Figure 4B); 37 genes were up-regulated and 47 genes were down-regulated in the Fructose group compared with the Lactose group (Figure 4C); 17 genes were up-regulated and 160 genes were down-regulated in the EPS group compared with the Control group (Figure 4D); 18 genes were up-regulated and 189 genes were down-regulated in the Protein group compared with the Control group (Figure 4E); 26 genes were up-regulated and 22 genes were down-regulated in the EPS group compared with the Protein group (Figure 4F).

Detailed gene expression profiles of the chromosome, plasmid 1, and plasmid 2 across four experimental groups, compared to the Control, were visualized using Circos plots (Figure 5). On the chromosome, these gene modules exhibit pronounced transcriptional regulation, including two distinct sets of prophage gene clusters, phosphotransferase systems (PTS) transporters, ATP-binding cassette (ABC) transporters, pathways related to carbohydrate metabolism, and a purine biosynthesis cluster (*Pur*) (Figure 5A). On the plasmids, genes associated with nucleotide biosynthesis, stress response, and sugar transport are prominently regulated (Figure 5B,C).

### 3.4. Significantly Regulated Transmembrane and Intracellular Transport Clusters

Compared to the control, the regulation of transmembrane and intracellular transport clusters in the other four groups shows significant variations, with notable up-regulation and down-regulation patterns observed in specific groups (Figure 6). The sucrose-specific PTS transporter ScrA was dramatically up-regulated in the Protein group (7.92 log_2_ fold change), whereas no dramatic changes were observed in the other three groups (Figure 6A). Glucose PTS transporter BglF was significantly down-regulated in all four groups (Figure 6B). PTS sugar transporter AgaBCDF was up-regulated in the Fructose groups but slightly down-regulated in Lactose group (Figure 6C). Glucitol/sorbitol PTS transporter SrlABE was up-regulated in the Fructose groups and Lactose group (Figure 6D). Mannose/fructose/sorbose PTS transporter ManXYZ was significantly down-regulated in Fructose group (Figure 6E). Galactitol PTS transporter GatABC was up-regulated in Fructose group (Figure 6F). As for ABC transporters, galactose oligomer/maltoligosaccharide ABC transporter GanOPQ-MsmX was only up-regulated in Lactose group (Figure 6G), while phosphate ABC transporter PstSCAB was down-regulated in Lactose and Protein groups (Figure 6H).

### 3.5. Significantly Regulated Pathways and Functions

Significant regulation of carbohydrate metabolic pathways was observed across the different experimental groups. For instance, fructose PTS transporters (FruA, FruB, and FruAb) were up-regulated in the Fructose group, while the intracellular fructokinase Sack1 showed the highest up-regulation in the Protein group (Figure 7A). Additionally, both 1-phosphofructokinase FruK and sorbitol-6-phosphate 2-dehydrogenase SrlD were both up-regulated in the Fructose group. Also, the lactose-galactose metabolism pathway, including genes such as beta-galactosidase LacZ, galactose mutarotase GalM, galactokinase GalK, galactose-1-phosphate uridylyltransferase GalT, and UDP-glucose 4-epimerase GalE, all exhibited dramatic up-regulation (Figure 7B) in the Lactose group. Furthermore, genes involved in the conversion among stachyose, mannotriose, raffinose, melibiose, sucrose, galactose, glucose, and fructose were regulated, with alpha-galactosidase GalA up-regulated in the Lactose group, while sucrose-6-phosphate hydrolase SacA and oligo-1,6-glucosidase MalL were both significantly up-regulated in the Protein group (Figure 7C).

The gene cluster responsible for the synthesis of inosinic acid from phosphoribosyl pyrophosphate and L-glutamine, a key component in the *de novo* purine biosynthesis pathway, was significantly down-regulated in the Fructose, Lactose, and Protein groups, with the highest down-regulation observed in the Fructose and Protein groups (Figure 8A). Regarding plasmid gene regulation, the ribonucleotide-diphosphate reductase nrdAB on plasmid 1 was slightly up-regulated in the Fructose group but showed a slight down-regulation in the other three groups (Figure 8B). The glycine betaine/L-proline ABC transporter ProVWX on plasmid 1 exhibited relatively stable yet high transcription activity across all groups, without significant regulation (Figure 8C). A couple of OsmC family proteins on plasmid 2 were dramatically up-regulated in the Fructose group but down-regulated in the EPS and Protein groups (Figure 8D). Lastly, a group of non-specific PTS sugar transporter subunits on plasmid 2 were up-regulated in the Fructose group but significantly down-regulated in the Protein group (Figure 8E).

## 4. Discussion

Understanding the genomic and transcriptomic basis of food microbial strains, even in well-characterized species, is essential for advancing applications, particularly in sectors emphasizing food safety and functional properties [26,27]. These studies provide insights into microbial genetics and gene expression, particularly in relation to metabolism, resistance, and ecological adaptation [28,29]. *L. plantarum* NMGL2, a strain isolated from traditional fermented dairy by our research group, has demonstrated remarkable bioactivities and potential food applications. Study showed this strain produces antimicrobial peptide (AMP) with potent antibacterial properties against various Gram-positive and Gram-negative pathogens [6]. The AMP was verified as structurally distinct and exhibits stability under a wide range of pH and temperature conditions. Furthermore, *L. plantarum* NMGL2 showed robust resistance to cold and acid stresses, with proteomic and metabolomic analyses revealing the activation of cellular mechanisms, such as enhanced biosynthesis of glycolipids, glycoproteins, and cell wall components, to counteract harsh environmental conditions [7,8]. These adaptations highlight the potential for improving the survivability of probiotics in fermented foods and other applications. Additionally, the EPS demonstrate a protective effect against inflammatory bowel disease (IBD) by regulating oxidative stress and inflammatory pathways, particularly through the suppression of NF-κB signaling and enhancement of intestinal barrier integrity [9]. Together, these findings position *L. plantarum* NMGL2 as a strain for developing functional food products.

Previous investigations of *L. plantarum* have provided valuable insights into sugar metabolism and gene regulation. Lu and others revealed the role of CcpA in carbon catabolite repression when cells were grown on glucose or fructooligosaccharides [30], while Cui and others analyzed carbohydrate metabolism genes and two-component systems in *L. plantarum* [31]. Zhao and others annotated EPS biosynthetic gene clusters in *L. plantarum* MC5 [32]. In our study, the WGS and comprehensive annotation of the *L. plantarum* NMGL2 genome, facilitated by publicly available databases and tools, offers new insights in understanding the metabolic and functional capabilities of this strain. WGS of *L. plantarum* NMGL2 provided a complete genomic landscape, including a detailed chromosomal and plasmid-based gene map. By employing the PGAP annotation pipeline, 3320 genes total genes were identified, including both CDS and RNA genes, which are fundamental to understanding the metabolic potential of this strain. Furthermore, KEGG-based functional annotation revealed key pathways associated with carbohydrate metabolism, signaling, and cellular processes. The documentation of this genomic data will serve as a valuable reference for future studies aiming to exploit *L. plantarum* NMGL2 in industrial applications such as fermented food and probiotic utilizations, also enabling precise genetic level engineering in future studies. Moreover, by using the strain-specific genome as a reference, we were able to conduct high-resolution transcriptomic studies at the nucleotide level. This approach not only allowed precise identification of individual gene transcription activity but also enabled accurate analysis of gene expression from the two plasmids (CP172592.1 and CP172593.1). Such detailed insights into plasmid-based transcriptional regulation are unattainable when using non-strain-specific reference genomes, underscoring the critical importance of having a genomic-transcriptomic analytical pipeline for comprehensive functional analyses.

Here, the growth study explored the effects of different carbon sources on EPS production, alongside bacterial growth dynamics. The results indicate that while the addition of fructose and lactose did not significantly increase EPS yield, the incorporation of plant-based soybean proteins led to a marked increase in EPS production, which could be linked to increased total bacterial population. This finding suggests that the EPS yield in *L. plantarum* NMGL2 is not solely dependent on carbohydrate nutrients but is also influenced by the nitrogen and protein content in the growth medium. This observation underscores the complexity of microbial metabolism, where protein supplementation appears to provide more favorable conditions for microbial EPS synthesis [33]. Also, the growth curves of the various groups mirrored the EPS production patterns, highlighting the interplay between bacterial growth and metabolic output. These findings are consistent with studies showing that a preferred nitrogen source can enhance EPS production by facilitating optimal growth conditions and stimulating specific metabolic pathways in LAB [34]. From an application perspective, these results suggest that adjusting the type and level of nitrogen sources may serve as a practical means to fine-tune microbial metabolism and enhance EPS productivity in industrial fermentations. The higher EPS yield observed with soybean proteins highlights the potential of plant-based protein supplements to improve the rheological and textural properties of fermented products. Such an approach could be particularly valuable for dairy and plant-derived fermentations, where optimized EPS production contributes not only to product quality and stability but also to the development of functional foods with added health benefits. The observed differences in growth performance, pH decline, and EPS yield under various culture conditions highlight the distinct physiological responses of *L. plantarum* NMGL2 to different nutrient environments. Faster acidification in the lactose and glucose groups suggests enhanced carbohydrate metabolism, potentially triggering acid stress adaptation mechanisms, while the protein group’s higher biomass and EPS yield indicate improved metabolic activity supported by a richer nitrogen source.

While fructose and lactose are well-documented carbohydrates that support the growth of LAB, their addition did not significantly enhance EPS production in *L. plantarum* NMGL2 [35,36]. However, their presence did induce substantial changes in the regulation of genes associated with sugar metabolism, particularly those involved in the uptake and utilization of carbohydrates (Figure 6 and Figure 7). This suggests that although these carbon sources may not directly promote EPS biosynthesis, they can modulate the microbial metabolic pathways to optimize sugar utilization. Similar findings have been reported in other LAB strains, where carbohydrate availability influences the expression of genes linked to carbohydrate metabolism, including transporters and enzymes involved in the breakdown and conversion of sugars [30]. The inclusion of EPS and plant-based soybean peptone in the culture medium significantly altered the transcriptomic profile of *L. plantarum* NMGL2. Compared to glucose control, both the EPS and protein groups showed broad transcriptional down-regulation, suggesting a potential feedback inhibition mechanism in response to the addition of these macromolecules. Interestingly, the EPS group displayed transcriptional patterns resembling those of the Protein group, despite the distinct nature of the added compounds. This observation suggests that the presence of large extracellular molecules, such as EPS or proteins, may create environmental changes that trigger broader transcriptional adjustments. In particular, the pronounced down-regulation of two prophage gene clusters could be a major factor contributing to the overall transcriptional similarity between the EPS and Protein groups (Figure 5A), reflecting a potential transcriptional change under these conditions. These effects are likely part of a broader physiological adjustment rather than a targeted regulatory response, though this possibility requires further verification. However, despite these similarities, detailed gene expression analysis revealed distinct differences in how each additive modulated specific metabolic pathways, such as those involved in carbohydrate and purine metabolism (Figure 7 and Figure 8A), highlight the essential role of nitrogen source and EPS accumulation in modulating microbial physiology.

The down-regulation of the two prophage gene clusters were observed in all four culture groups when compared with control (Figure 5A). Prophages are integrated viral elements within bacterial genomes that can influence host metabolism, stress response, and even contribute to horizontal gene transfer [37]. Also, prophage analysis across diverse LAB species revealed widespread presence and high diversity and have been verified to broadly exist in fermented foods [38,39,40]. Antibiotic resistance genes were commonly associated with prophages, potentially influencing bacterial resistance, while CRISPR-Cas systems were identified as possible antagonistic elements against prophages. Previous studies have shown that stressors, such as low pH, high temperature, and NaCl, induced minimal or no prophage activation in *Lactococcus lactis* [41]. In this study, the observed down-regulation of prophage-related genes, particularly under EPS supplementation and protein-rich conditions, suggests a potential association between environmental factors and prophage gene expression. However, the current evidence is correlative rather than mechanistic, and further investigation of the prophage genomic context and possible regulatory elements is needed to better understand this relationship.

Moreover, PTS and ABC transporters were the most significantly regulated genes in *L. plantarum* NMGL2 (Figure 6). Such transporters are crucial for the uptake of carbohydrates, and their regulation highlights the adaptive nature of LAB to varying nutritional and stress environments [31]. The PTS transporters, known for their high specificity and energy efficiency, play a critical role in controlling sugar uptake in LAB. Significant regulation of carbohydrate metabolic pathways was observed across all experimental groups. In the fructose group, fructose-specific PTS transporters and fructokinase were up-regulated, suggesting an enhanced capacity for utilization of fructose. Similarly, in the lactose group, genes involved in lactose and galactose metabolism were up-regulated, indicating a shift toward the utilization of lactose as a primary carbon source. The synergistic interactions between probiotics and prebiotics have emerged as a focal point in research, driven by their pivotal role in modulating gut health and host physiology [42]. The conversion pathways of oligosaccharides and monosaccharides (Figure 7C) in *L. plantarum* showcase its metabolic flexibility, highlighting interactions with carbohydrates of varying polymerization degrees, which hold potential for probiotic and prebiotic applications.

LAB plasmids are extrachromosomal DNA that enhance metabolic adaptability, stress resistance, and bacteriocin production, playing a vital role in fermentation, probiotics, and biotechnology [43]. The genomic and transcriptomic analyses revealed the significant role of two plasmids in the metabolic regulation of *L. plantarum* NMGL2. Both plasmids exhibited distinct regulatory profiles, with plasmid 1 showing up-regulation in genes involved in nucleotide biosynthesis, stress responses, and sugar transport, and plasmid 2 influencing transport and osmotic stress regulation. These plasmids may confer specific metabolic advantages, such as enhanced tolerance to environmental stresses or the ability to utilize diverse carbon sources. The ability to study both genomic and transcriptomic responses in parallel provides valuable insights into the functional importance of plasmids in LAB strains and underscores the need for plasmid-related strategies in the control of LAB for industrial applications.

## 5. Conclusions

This study provides a comprehensive genomic, metabolic, and transcriptomic analysis of *L. plantarum* NMGL2, highlighting its adaptability to diverse nutritional environments. Using high-throughput sequencing and in silico tools, we constructed precise genomic and transcriptomic profile of the strain, unveiling key metabolic pathways involved in carbohydrate metabolism, nutritional adaptation, and bioactive compound production. These findings extend beyond stress response of this strain, revealing how environmental nutrients reprogram gene expression to optimize general metabolism and bioactive compound synthesis. The strain’s response to various carbon sources, along with the impact of EPS supplementation and soy peptone substitution, demonstrated the intricate mechanism between nitrogen and carbohydrate sources in regulating microbial metabolism. Moreover, the down-regulation of prophage gene clusters under experimental conditions, alongside the regulation of transporters and carbohydrate pathways, suggests actively regulated microbial mechanisms that may influence the strain’s stability and performance in the fermentation processes. Overall, these findings provide valuable insights for developing novel probiotic applications and lay a foundation for future metabolic engineering strategies in LAB strains.

## Figures and Tables

**Figure 1 foods-14-03520-f001:**
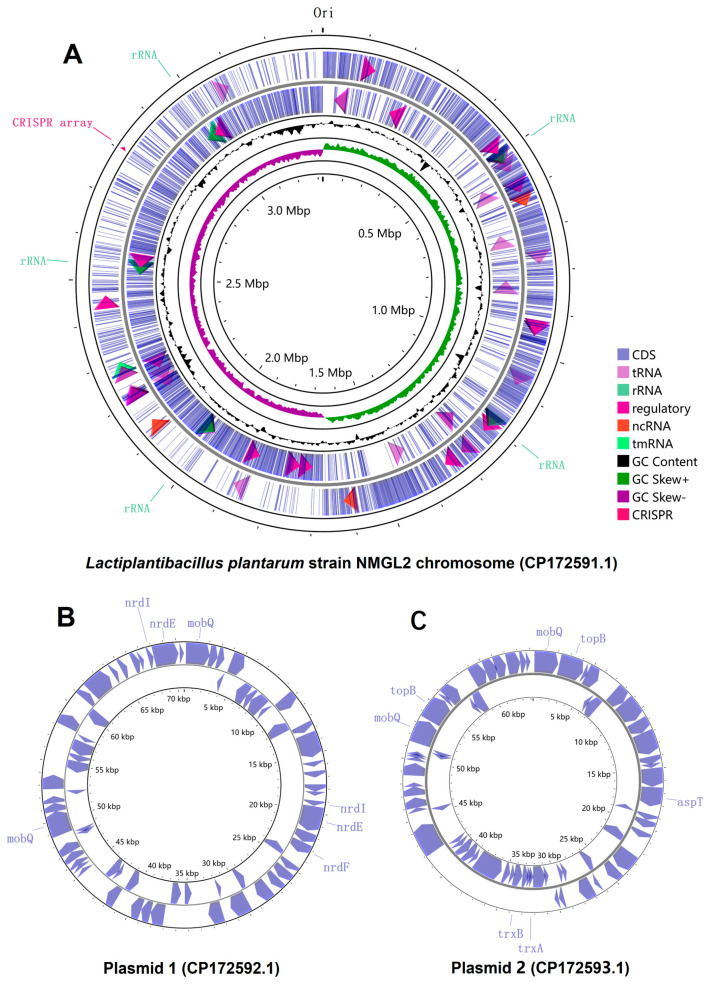
Circular genome depiction of *L. plantarum* NMGL2. (**A**) Chromosome, (**B**) Plasmid 1, and (**C**) Plasmid 2. Genes are displayed according to transcriptional orientation, with counterclockwise transcription shown on the inner circle and clockwise transcription on the outer circle.

**Figure 2 foods-14-03520-f002:**
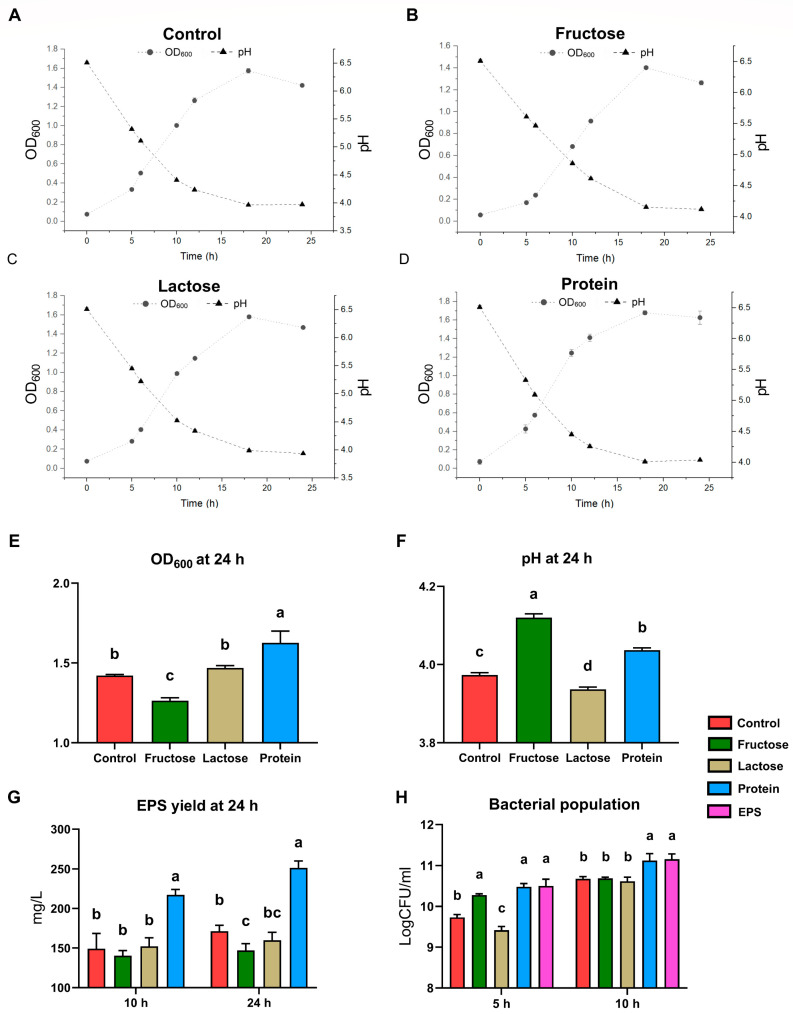
Culture dynamics of *L. plantarum* NMGL2. Culture optical density measured at the wavelength of 600 nm (OD_600_), pH value, and EPS yield of (**A**) Control group, (**B**) Fructose group, (**C**) Lactose group, and (**D**) Protein group. (**E**) OD_600_, (**F**) pH value, and (**G**) EPS yield of four groups without exogenous EPS supplementation at 24 h. (**H**) Bacterial population of all five groups determined using plate counting method at 5 h and 10 h. Different lowercase letters indicate statistically significant differences (*p* < 0.05) among groups sampled at the same time point.

**Figure 3 foods-14-03520-f003:**
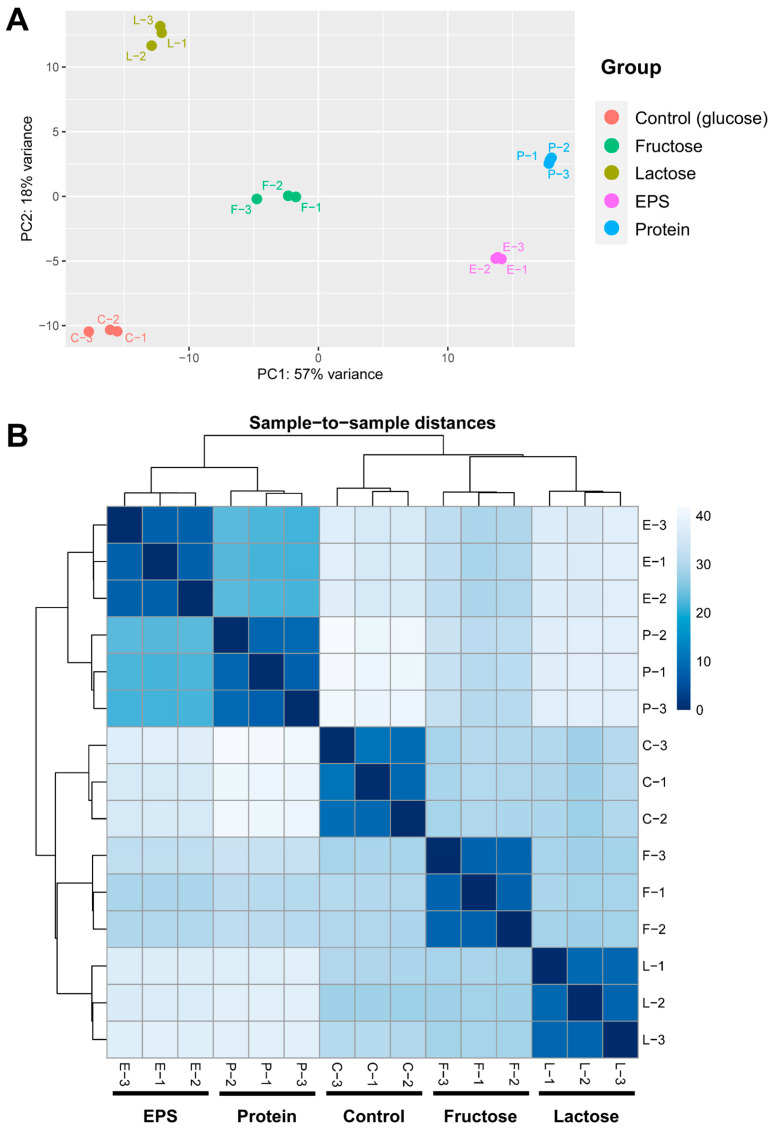
RNA-seq data assessment. (**A**) PCA plot illustration of the clustering of five groups based on normalized htseq-counts. (**B**) Heatmap displaying the pairwise sample-to-sample distances in gene expression across 15 individual samples.

**Figure 4 foods-14-03520-f004:**
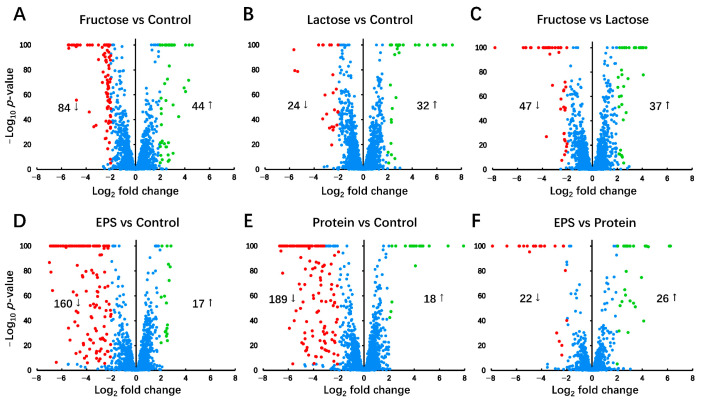
Global gene expression analysis. Volcano plot of normalized gene expression levels between each group. (**A**) Fructose vs. Control, (**B**) Lactose vs. Control, (**C**) Fructose vs. Lactose, (**D**) EPS vs. Control, (**E**) Protein vs. Control, and (**F**) EPS vs. Protein. Genes up-regulated in the former group relative to the latter group are presented as positive log_2_ fold change values, whereas down-regulated genes are presented as negative log_2_ fold change values. Genes with significantly higher expression (log_2_ fold change > 2 and -log_10_ *p*-value > 5) are labeled in green, while those with significantly lower expression (log_2_ fold change < −2 and -log_10_ *p*-value > 5) are labeled in red. Genes with a −log_10_ *p*-value greater than 100 were capped at 100 to ensure consistent scaling in the volcano plots.

**Figure 5 foods-14-03520-f005:**
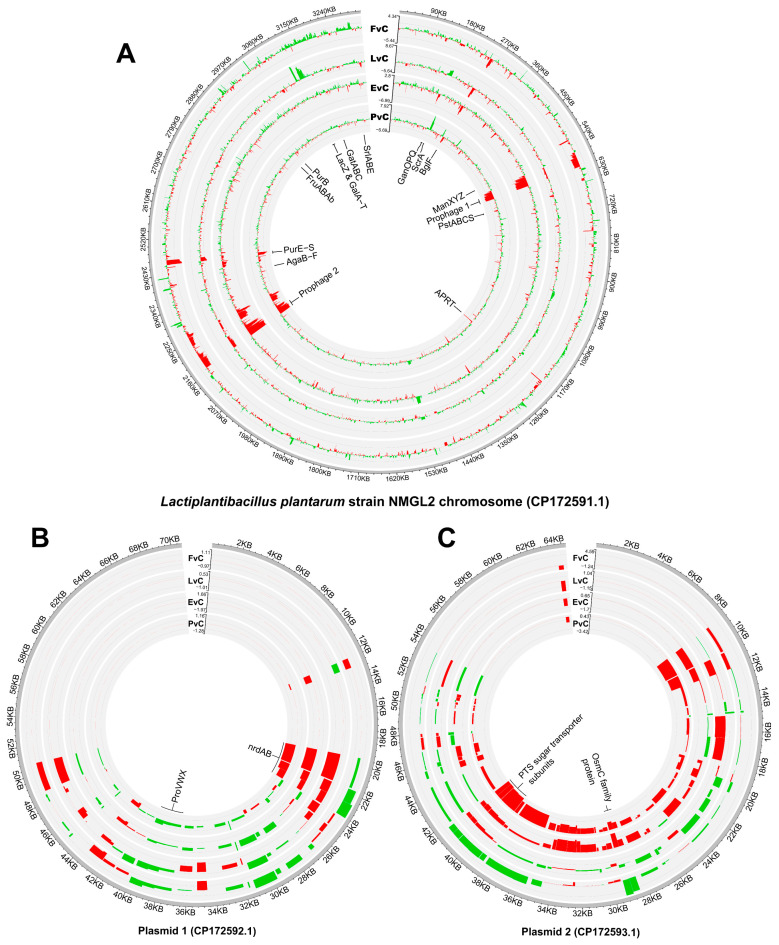
Circos plot illustration of global transcriptional changes between paired group comparisons. (**A**) Chromosome, (**B**) Plasmid 1, and (**C**) Plasmid 2. Positive values with green bars representing higher gene expression in the former group while negative values with red bars representing higher gene expression in the latter group. From outer to inner circles, FvC: Fructose vs. Control, LvC: Lactose vs. Control, EvC: EPS vs. Control, PvC, Protein vs. Control. The radial axis reflects the range of expression differences.

**Figure 6 foods-14-03520-f006:**
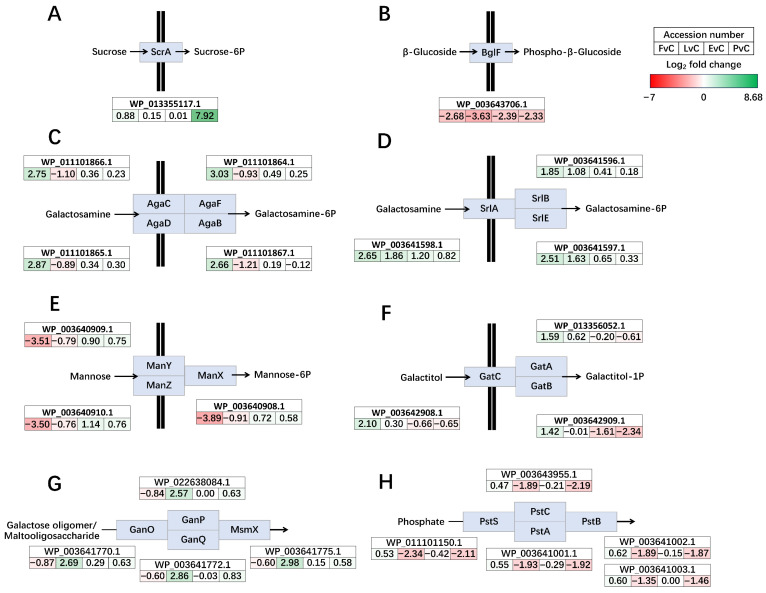
Significantly regulated phosphotransferase systems (PTS) transporters (**A**–**F**) and ATP-binding cassette (**A**–**C**) transporters (**G**,**H**). (**A**) Sucrose-specific PTS transporter ScrA, (**B**) PTS glucose transporter BglF, (**C**) PTS sugar transporter AgaBCDF, (**D**) PTS glucitol/sorbitol transporter SrlABE, (**E**) PTS mannose/fructose/sorbose transporter ManXYZ, (**F**) PTS galactitol transporter GatABC, (**G**) carbohydrate ABC transporter GanOPQ-MsmX, and (**H**) phosphate ABC transporter PstSCAB. Genes up-regulated in the former group are shown in green, and those down-regulated are shown in red. The color intensity reflects the magnitude of differential expression. FvC: Fructose vs. Control, LvC: Lactose vs. Control, EvC: EPS vs. Control, PvC, Protein vs. Control.

**Figure 7 foods-14-03520-f007:**
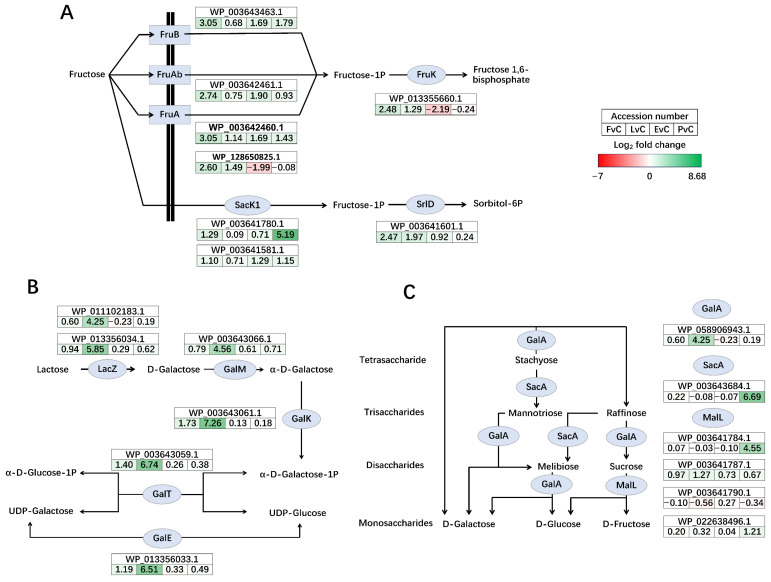
Significantly regulated carbohydrate metabolism pathways. (**A**) Fructose cross membrane transportation, phosphorylation, and conversion to sorbitol, (**B**) Lactose-galactose metabolism pathway, and (**C**) metabolism and conversion of mono-, di-, tri-, and tetrasaccharides. Genes up-regulated in the former group are shown in green, and those down-regulated are shown in red. The color intensity reflects the magnitude of differential expression. FvC: Fructose vs. Control, LvC: Lactose vs. Control, EvC: EPS vs. Control, PvC, Protein vs. Control.

**Figure 8 foods-14-03520-f008:**
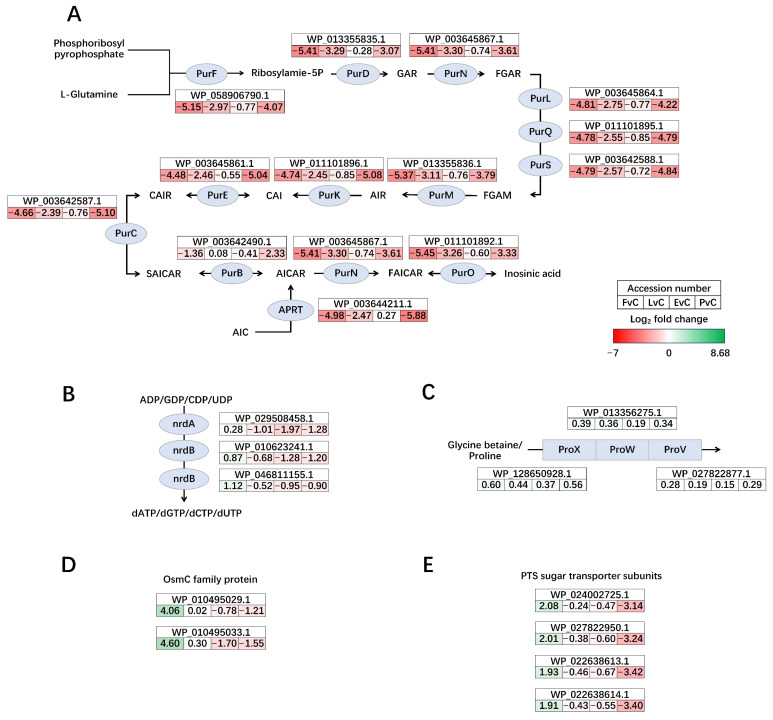
Significantly regulated gene clusters of *de novo* purine biosynthesis (**A**) and on plasmids (**B**–**E**). (**A**) The conversion pathway of phosphoribosyl pyrophosphate and glutamine into inosine acid, (**B**) ribonucleotide-diphosphate reductase nrdAB on plasmid 1, (**C**) glycine betaine/L-proline ABC transporter ProVWX on plasmid 1, (**D**) OsmC family proteins on plasmid 2, and (**E**) PTS sugar transporter subunits on plasmid 2. Genes upregulated in the former group are shown in green, and those downregulated are shown in red. The color intensity reflects the magnitude of differential expression. FvC: Fructose vs. Control, LvC: Lactose vs. Control, EvC: EPS vs. Control, PvC, Protein vs. Control.

**Table 1 foods-14-03520-t001:** Detailed composition of the culture media for the five experimental groups.

Component	Group Name (Denotation)				
	Control (C)	Fructose (F)	Lactose (L)	EPS (E)	Protein (P)
Monosaccharide or disaccharide	Glucose 15 g/L	Fructose 15 g/L	Lactose 15 g/L	Glucose 15 g/L	Glucose 15 g/L
Peptone	Beef tryptic digest 10 g/L	Beef tryptic digest 10 g/L	Beef tryptic digest 10 g/L	Beef tryptic digest 10 g/L	Soybean papaic digest 30 g/L
Freeze-dried EPS by *Lactiplantibacillus plantarum* NMGL2	-	-	-	10 g/L	-
Other shared ingredients	Yeast extract 4 g/L, beef extract 5 g/L, sodium acetate 5 g/L, magnesium sulfate 0.2 g/L, manganese sulfate 0.05 g/L, dipotassium hydrogen phosphate 2 g/L, triammonium citrate 2 g/L, Tween 80 1 mL				

## Data Availability

The assembled and annotated bacterial genome was deposited into NCBI BioProject database under the accession number of PRJNA1178815. Paired-end raw RNA sequencing data in FASTQ format was deposited into NCBI BioProject database under accession number of PRJNA1176381. Other data will be made available on request.

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
