# Peer review of "Genomic and Transcriptomic Dissection of Growth Characteristics and Exopolysaccharide-Related Bioactivities in Lactiplantibacillus plantarum NMGL2"

_foods, 2025, doi:10.3390/foods14203520_

Round 1
Reviewer 1 Report
Comments and Suggestions for Authors
The manuscript presents a comprehensive genomic and transcriptomic analysis of Lactiplantibacillus plantarum NMGL2, focusing on growth dynamics, EPS (exopolysaccharide) production, and metabolic regulation under different nutritional conditions. Overall, the study is well designed, the data appear robust, and the manuscript is clearly structured. However, several areas require clarification, additional detail, and refinement to improve readability and scientific rigor.
-
The manuscript emphasizes NMGL2’s EPS-related bioactivities, but many previous studies from the same group have already reported on its antimicrobial peptides, EPS production, and stress resistance (see references [6–9]). The authors should explicitly state what new genomic and transcriptomic insights are provided here beyond their earlier proteomic and metabolomic analyses.
-
Please clarify the novelty of this work in comparison with other L. plantarum transcriptomic studies, particularly those focusing on sugar metabolism and EPS biosynthesis.
-
Why was EPS supplementation set at 10 g/L? Is this concentration physiologically or industrially relevant? Please justify this choice.
-
RNA was collected only at 10 h. The rationale for selecting this single time point should be elaborated, especially since EPS yield was monitored at both 10 h and 24 h. Could transcriptomic changes at later stages differ significantly?
-
The downregulation of prophage gene clusters is an important finding; however, the discussion remains somewhat speculative. The authors should either provide supporting data (e.g., prophage annotation, genomic context, or possible stress-response elements) or moderate their conclusions.
-
The similarity between EPS and protein groups is intriguing but underexplored. Could osmotic stress or viscosity-related effects account for this observation? Please expand the discussion.
-
Figures 2–8 are data-rich but sometimes difficult to interpret. The legends should clearly indicate which groups are being compared and what the colors represent. For example, in Figure 4 (volcano plots), the cutoff thresholds should be explicitly mentioned in the legend.
-
The transcriptomic data require validation. Please include a qRT-PCR experiment to support the RNA-seq results.
Author Response
The manuscript presents a comprehensive genomic and transcriptomic analysis of Lactiplantibacillus plantarum NMGL2, focusing on growth dynamics, EPS (exopolysaccharide) production, and metabolic regulation under different nutritional conditions. Overall, the study is well designed, the data appear robust, and the manuscript is clearly structured. However, several areas require clarification, additional detail, and refinement to improve readability and scientific rigor.
Response: We sincerely thank the reviewer for the comment. We have carefully addressed all suggested points to improve clarity, detail, and scientific rigor in the revised manuscript.
1.The manuscript emphasizes NMGL2’s EPS-related bioactivities, but many previous studies from the same group have already reported on its antimicrobial peptides, EPS production, and stress resistance (see references [6–9]). The authors should explicitly state what new genomic and transcriptomic insights are provided here beyond their earlier proteomic and metabolomic analyses.
Response: We sincerely thank the reviewer for this insightful comment. Our previous studies mainly focused on the in vitro physiological adaptation of L. plantarum NMGL2 under cold and acid stresses, where proteomic and metabolomic analyses revealed enhanced glycolipid and cell wall biosynthesis, supporting stress resistance and cell integrity during fermented milk storage. The previous works have been addressed in the first paragraph of Discussion section. In contrast, the present work extends beyond stress response to explore the genomic and transcriptomic basis of EPS production and nutrient regulation. By integrating whole-genome sequencing with RNA-seq under different carbon (glucose, fructose, lactose, and EPS addition) and nitrogen (soy peptone) conditions, we identified key regulatory pathways and transporter systems that link nutrient utilization to overall metabolism.
As recommended, we have added a sentence in the Conclusion section to highlight the novelty of this work. The added sentence is following: “These findings extend beyond stress response of this strain, revealing how environmental nutrients reprogram gene expression to optimize general metabolism and bioactive compound synthesis.”
2.Please clarify the novelty of this work in comparison with other L. plantarum transcriptomic studies, particularly those focusing on sugar metabolism and EPS biosynthesis.
Response: Thank you Previous L. plantarum transcriptomic studies have mainly focused on aspects of sugar metabolism or gene regulation. For example, Lu et al. (2018) examined CcpA-dependent transcriptional regulation using FOS or glucose as carbon sources, revealing carbon catabolite repression. Cui et al. (2021) characterized strain-level variation in carbohydrate metabolism genes and analyzed expression of several sugar metabolism and TCS genes by qRT-PCR. Zhao et al. (2023) performed comparative genomic annotation of EPS biosynthesis clusters in strain MC5 using WGS.
In contrast, our study integrates complete genome sequencing with multi-condition RNA-seq, providing a comprehensive view of how different carbon and nitrogen sources reprogram global and plasmid-level transcription in L. plantarum NMGL2. We identified coordinated regulation among PTS/ABC transporters, purine metabolism, and EPS biosynthetic genes. The following paragraph is added at the beginning of 2nd paragraph of Discussion section to address these findings: “Previous investigations of L. plantarum have provided valuable insights into sugar metabolism and gene regulation. Lu and others revealed the role of CcpA in carbon catab-olite repression when cells were grown on glucose or fructooligosaccharides [31], while Cui and others analyzed carbohydrate metabolism genes and two-component systems in L. plantarum [32]. Zhao and others annotated EPS biosynthetic gene clusters in L. plantarum MC5 [33].”
3.Why was EPS supplementation set at 10 g/L? Is this concentration physiologically or industrially relevant? Please justify this choice.
Response: Thank you for the question. We chose 10 g/L EPS because this concentration is analogous to the ~1% sugar supplementation commonly used in microbial culture media—it provides a meaningful carbohydrate representation input without overly perturbing osmotic balance. As a secreted polymer, EPS at this level may influence secretion equilibrium, but this concentration is within the range reported in high-yield EPS fermentations (for example, some strains can reach several g/L to around 10 g/L under optimized conditions) (Elmansy et al., 2023, Liu et al., 2016). In future work, we will examine a wider range of EPS concentrations to dissect mechanistic and industrial relevance.
4.RNA was collected only at 10 h. The rationale for selecting this single time point should be elaborated, especially since EPS yield was monitored at both 10 h and 24 h. Could transcriptomic changes at later stages differ significantly?
Response: We appreciate the reviewer’s thoughtful comment. RNA samples were collected at 10 h, corresponding to the exponential phase, when cell density (OD₆₀₀ ≈ 10) was sufficient for reliable RNA extraction and when cells exhibited active metabolism and EPS synthesis. In contrast, 24 h represents the stationary phase, dominated by nutrient depletion and stress responses rather than primary metabolism. Therefore, sampling at 10 h allowed us to capture transcriptional profiles most representative of vigorous growth and EPS production. We agree that later-stage transcriptomic changes could provide additional insights, and future studies focusing on fermented food applications or bioactive compound production will include multi-timepoint analyses to explore transcriptional regulation dynamics.
5.The downregulation of prophage gene clusters is an important finding; however, the discussion remains somewhat speculative. The authors should either provide supporting data (e.g., prophage annotation, genomic context, or possible stress-response elements) or moderate their conclusions.
Response: We thank the reviewer for this valuable suggestion. We have moderated the interpretation regarding prophage regulation and revised the discussion to emphasize correlation rather than causation. The revision paragraph in Discussion section is following: “In this study, the observed down-regulation of prophage-related genes, particularly under EPS supplementation and protein-rich conditions, suggests a potential association between environmental factors and prophage gene expression. However, the current evidence is correlative rather than mechanistic, and further investigation of the prophage genomic context and possible regulatory elements is needed to better understand this relationship.”
6.The similarity between EPS and protein groups is intriguing but underexplored. Could osmotic stress or viscosity-related effects account for this observation? Please expand the discussion.
Response: Thank you. We have revised the discussion to present a more balanced interpretation. The similarity in transcriptional responses between the EPS and protein groups may be partly influenced by changes in extracellular conditions, such as osmotic pressure, viscosity, or molecular crowding, caused by the presence of large macromolecules. These factors could subtly affect global gene regulation, potentially including the coordinated down-regulation of prophage-related genes. Given that prophage clusters occupy a considerable portion of the L. plantarum NMGL2 genome, such transcriptional trends might reflect a general adjustment to altered environmental conditions rather than a specific regulatory mechanism. Further investigation will be needed to confirm this possibility.
The revised discussion is following: “Interestingly, the EPS group displayed transcriptional patterns resembling those of the protein group, despite the distinct nature of the added compounds. This observation suggests that the presence of large extracellular molecules, such as EPS or proteins, may create environmental changes that trigger broader transcriptional changes. In particular, the pronounced down-regulation of two prophage gene clusters could be a major factor contributing to the overall transcriptional similarity between the EPS and protein groups (Figure 5A), reflecting a potential transcriptional change under these conditions. These effects are likely part of a broader physiological adjustment rather than a targeted regulatory response, though this possibility requires further verification.”
7.Figures 2–8 are data-rich but sometimes difficult to interpret. The legends should clearly indicate which groups are being compared and what the colors represent. For example, in Figure 4 (volcano plots), the cutoff thresholds should be explicitly mentioned in the legend.
Response: We thank the reviewer for this constructive suggestion. Figure legends have been revised for improvement. For Figure 4 (volcano plots), the cutoff thresholds have been explicitly stated in the legend: “Genes with a -log₁₀ p-value greater than 100 were capped at 100 to ensure consistent scaling in the volcano plots.”
8.The transcriptomic data require validation. Please include a qRT-PCR experiment to support the RNA-seq results.
Response: We thank the reviewer for this important comment. We agree that transcriptomic validation is important especially when the adopted pipeline is not validated. However, our previous studies using the same RNA-seq analytical pipeline and Illumina HiSeq platform have already demonstrated excellent consistency between RNA-seq and qRT-PCR results in lactic acid bacteria (LAB). For example, one of our research (Han et al., 2021a) validated six genes by qRT-PCR and observed nearly identical expression pattern between RNA-seq and qRT-PCR, while another our study (Han et al., 2021b) validated 15 genes and found no meaningful deviation between qRT-PCR and RNA-seq results, but also demonstrated that qRT-PCR could generate additional information under conditions or time points that were not covered by RNA-seq analysis. In addition, several recent studies on LAB have adopted Illumina-only validation, given that the high sequencing depth, biological replicates, and robust in silico pipeline ensure reproducibility (Liu et al., 2024, Zhai et al., 2020). These existing studies, together with our results, PCA and heatmap analyses in particular, suggest that the Illumina-based RNA-seq data in this study provide robust and reliable quantitative insights under current analytical standards, indicating that additional qRT-PCR validation may not be strictly necessary. Nevertheless, we fully acknowledge the reviewer’s point. In future targeted studies, especially those focusing on different conditions and/or specific gene clusters (e.g., EPS biosynthesis or regulation mechanisms), we plan to include qRT-PCR confirmation as part of validation and exploration strategy. This approach will not only be economically and scientifically sound but also provide more detailed insights.
References:
- Lu, Y., Song, S., Tian, H., Yu, H., Zhao, J., & Chen, C. (2018). Functional analysis of the role of CcpA in Lactobacillus plantarum grown on fructooligosaccharides or glucose: a transcriptomic perspective. Microbial Cell Factories, 17(1), 201.
- Cui, Y., Wang, M., Zheng, Y., Miao, K., & Qu, X. (2021). The carbohydrate metabolism of Lactiplantibacillus plantarum. International Journal of Molecular Sciences, 22(24), 13452.
- Zhao, X., Liang, Q., Song, X., & Zhang, Y. (2023). Whole genome sequence of Lactiplantibacillus plantarum MC5 and comparative analysis of eps gene clusters. Frontiers in Microbiology, 14, 1146566.
- Elmansy, E. A., Elkady, E. M., Asker, M. S., Abdallah, N. A., Khalil, B. E., & Amer, S. K. (2023). Improved production of Lactiplantibacillus plantarum RO30 exopolysaccharide (REPS) by optimization of process parameters through statistical experimental designs. BMC microbiology, 23(1), 361.
- Liu, Q., Huang, X., Yang, D., Si, T., Pan, S., & Yang, F. (2016). Yield improvement of exopolysaccharides by screening of the Lactobacillus acidophilus ATCC and optimization of the fermentation and extraction conditions. EXCLI journal, 15, 119.
- Han, D., Shi, R., Yan, Q., Shi, Y., Ma, J., & Jiang, Z. (2021a). Global transcriptomic analysis of functional oligosaccharide metabolism in Pediococcus pentosaceus. Applied Microbiology and Biotechnology, 105(4), 1601-1614.
- Han, D., Yan, Q., Liu, J., Jiang, Z., & Yang, S. (2021b). Transcriptomic analysis of Pediococcus pentosaceus reveals carbohydrate metabolic dynamics under lactic acid stress. Frontiers in Microbiology, 12, 736411.
- Liu, X., Zhao, J., Zang, J., Peng, C., Lv, L., & Li, Z. (2024). Integrated analysis of physiology, antioxidant activity and transcriptomic of Lactobacillus plantarum 120 in response to acid stress. LWT, 214, 117109.
- Zhai, Z., Yang, Y., Wang, H., Wang, G., Ren, F., Li, Z., & Hao, Y. (2020). Global transcriptomic analysis of Lactobacillus plantarum CAUH2 in response to hydrogen peroxide stress. Food microbiology, 87, 103389.

Reviewer 2 Report
Comments and Suggestions for Authors
The article presents genomic and transcriptomic data will serve as a valuable reference for future studies aiming to exploit L. plantarum NMGL2 in industrial applications such as bioactive compound production and probiotic foods.
The article presents novel research and is relevant to the journal's audience. The introduction is clear and provides a good summary of the current state of the literature.
However, the objectives of the work should be clearly explained: (i) to conduct genomic and transcriptomic studies for better application of the strain? (ii) to promote EPS production under different conditions? Both?
What objectives are pursued with the different experimental conditions tested, and why are they chosen? For example, why are soy hydrolysates or EPS sources included?
There is also a certain lack of coherence between the results and the discussion and conclusions presented, especially regarding metabolism, resistance, ecological suitability, host-microbe interactions, and production of bioactive compounds, as these are not aspects addressed in the results.
Specific Comments
L 46 correct italics in L. plantarum
L 387 The authors state that "These studies reveal microbial genetics and gene expression, providing insights into metabolism, resistance, ecological fitness, and host-microbe interactions essential for dietary and therapeutic advances." Which of the presented results contribute to our understanding of the issues you mention in the discussion?
L 433: Which protein components of soybeans (compared to other nitrogen sources) do you think are responsible for promoting EPS production?
L 513: What stress resistance and bioactive compound production are evident in this study?
Author Response
The article presents genomic and transcriptomic data will serve as a valuable reference for future studies aiming to exploit L. plantarum NMGL2 in industrial applications such as bioactive compound production and probiotic foods. The article presents novel research and is relevant to the journal's audience. The introduction is clear and provides a good summary of the current state of the literature.
Response: We sincerely thank the reviewer for the recognition of our work and encouraging comments. Based on your valuable suggestions, we have carefully revised the manuscript to improve its clarity, depth, and overall quality.
However, the objectives of the work should be clearly explained: (i) to conduct genomic and transcriptomic studies for better application of the strain? (ii) to promote EPS production under different conditions? Both? What objectives are pursued with the different experimental conditions tested, and why are they chosen? For example, why are soy hydrolysates or EPS sources included?
Response: We appreciate the reviewer’s comment. The objectives of this work include both genomic and transcriptomic analyses of L. plantarum NMGL2 to better understand its potential applications and to explore factors influencing EPS production. The genomic and transcriptomic data provide a foundation of academic and practical value for future strain optimization. Although pure EPS production by L. plantarum is relatively uncommon, EPS plays an important role in fermented foods such as yogurt, cheese, fermented cereals, and probiotic formulations. Therefore, the different carbon sources and soy hydrolysates were selected to simulate potential fermentation environments, allowing us to examine how nutrient composition affects gene expression and EPS biosynthesis. Future studies will extend these findings to specific food systems and industrial applications.
The sentence in the last paragraph of Introduction section has now been modified to convey these objectives: “Through culture studies, whole genome sequencing, and transcriptomic analyses, we aim to investigate the metabolic characteristics of L. plantarum NMGL2 and identify the genes involved in the production of bioactive compounds, thereby supporting its potential applications in fermented foods and probiotic utilizations.”
There is also a certain lack of coherence between the results and the discussion and conclusions presented, especially regarding metabolism, resistance, ecological suitability, host-microbe interactions, and production of bioactive compounds, as these are not aspects addressed in the results.
Response: We thank the reviewer for this helpful comment. The introductory paragraph of the Discussion section has been revised to improve coherence with the presented Results and to more accurately reflect the study’s actual findings, particularly regarding metabolic regulation and related transcriptomic patterns. The revised sentence is following: “These studies provide insights into microbial genetics and gene expression, particularly in relation to metabolism, resistance, and ecological adaptation [28,29].”
Specific Comments
L 46 correct italics in L. plantarum
Response: Thank you. It has now been corrected.
L 387 The authors state that "These studies reveal microbial genetics and gene expression, providing insights into metabolism, resistance, ecological fitness, and host-microbe interactions essential for dietary and therapeutic advances." Which of the presented results contribute to our understanding of the issues you mention in the discussion?
Response: We thank the reviewer for this insightful comment. The mentioned sentence was originally intended as a general summary of previous research on related L. plantarum strains. In the revised version, this section has been substantially rewritten to focus on the findings and potential applications specific related to the present study. The updated statement now reads: “Together, these findings position L. plantarum NMGL2 as a strain for developing functional food products.”
L 433: Which protein components of soybeans (compared to other nitrogen sources) do you think are responsible for promoting EPS production?
Response: We used soybean hydrolysate at a higher concentration (30 g/L compared with 10 g/L beef tryptic digest) because it supplied more organic nitrogen, which may create metabolic conditions favorable for EPS synthesis and influence the overall balance between carbon and nitrogen utilization. In addition, the use of a soy-based nitrogen source better reflects plant-origin or food-related fermentation environments, such as cereal- or soy-based substrates, which are relevant to the potential applications of L. plantarum NMGL2.
L 513: What stress resistance and bioactive compound production are evident in this study?
Response: We thank the reviewer for this question. In this study, stress-related and bioactive compound–associated responses were mainly reflected in the differential effects of various carbon sources and EPS accumulations. For example, fermentation with lactose or glucose led to a faster pH decline and may cause the metabolic regulations to acid stress adaptation. At the same time, the accumulation of EPS may have contributed to maintaining osmotic balance under these conditions. Moreover, the transcriptomic data indicated changes in the expression of genes related to EPS and potential oligosaccharide metabolism, suggesting that these carbohydrate molecules could play roles in microbial adaptation, fermentation performance, and the formation of bioactive components relevant to food applications.

Reviewer 3 Report
Comments and Suggestions for Authors
The manuscript entitled “Genomic and Transcriptomic Dissection of Growth Characteristics and Exopolysaccharide-Related Bioactivities in Lactiplantibacillus plantarum NMGL2” provides valuable genomic and transcriptomic insights into L. plantarum NMGL2 with relevance for functional food development. The manuscript is scientifically sound and addresses an interesting topic for Foods. However, before acceptance, some revisions are necessary to improve clarity. Please find my detailed major and minor comments below.
Major Comments
- Lines 39–47: The introduction repeats the probiotic benefits of LABs and plantarum. Example: “LAB are valued for their probiotic properties… L. plantarum… known for its adaptability… and probiotic properties.” Condense to avoid redundancy. Suggested revision:“LAB are valued for their probiotic properties, including modulation of gut microbiota and production of beneficial metabolites. Among them, Lactiplantibacillus plantarum is widely studied for its adaptability and health-promoting effects.”
- Lines 83–92, 135–144, 169–171: Instruments and brand names are repeated excessively (e.g., “NanoDrop 2000 spectrophotometer (Thermo Fisher Scientific, Wilmington, DE, USA)”). List models/manufacturers only once. Example: “DNA purity was assessed using a NanoDrop spectrophotometer (Thermo Fisher).”
- Lines 183–189: Only ANOVA/Tukey is mentioned. For RNA-seq, DESeq2 usually applies multiple testing correction. Add: “For RNA-seq, differential expression analysis was conducted using DESeq2 with Benjamini–Hochberg correction for multiple testing (adjusted p < 0.05 considered significant).”
- Lines 427–440: The discussion repeats results (“protein supplementation led to higher EPS yield”) without deeper interpretation. Expand on industrial applications, e.g., how nitrogen source manipulation could improve dairy fermentation texture and EPS yields.
- English clarity
- Line 46: “...can facility unlocking...” → “...can facilitate unlocking...”
- Line 431: “EPS yield is not solely dependent on and regulated by carbohydrate nutrients...” → “EPS yield is not solely dependent on carbohydrate nutrients...”
Minor Comments
- Line 12: “Analyzing… via molecular and bioinformatics tools holds significant importance...”→ “Analyzing… using molecular and bioinformatics tools is important...”
- Line 54: “candidate for functional food development”→ “a promising probiotic candidate for functional food development.”
- Line 143: “The pH of each sample at each time point was measured...” → “The pH of samples was measured...”
- Line 254: “demonstrated a significantly higher population compared with the other three groups.” → “demonstrated significantly higher viable counts than the other three groups.”
- Line 407: “These adaptations underline its potential...” → “These adaptations highlight its potential...”
- Line 37: “Lactic acid bacteria (LAB) are a diverse group...” → (LABs)
- Line 39: “LAB has long been utilized in food production...” → LABs have long been utilized...
- References: Some journal names are not abbreviated in Foods style (e.g., “Food Bioscience” → “Food Biosci.”). Please check conformity.
Author Response
The manuscript entitled “Genomic and Transcriptomic Dissection of Growth Characteristics and Exopolysaccharide-Related Bioactivities in Lactiplantibacillus plantarum NMGL2” provides valuable genomic and transcriptomic insights into L. plantarum NMGL2 with relevance for functional food development. The manuscript is scientifically sound and addresses an interesting topic for Foods. However, before acceptance, some revisions are necessary to improve clarity. Please find my detailed major and minor comments below.
Response: We sincerely thank the reviewer for the assessment and constructive suggestions. We have carefully revised the manuscript to address all major and minor comments and improved clarity throughout. A detailed, point-by-point response is provided below.
Major Comments
Lines 39–47: The introduction repeats the probiotic benefits of LABs and plantarum. Example: “LAB are valued for their probiotic properties… L. plantarum… known for its adaptability… and probiotic properties.” Condense to avoid redundancy. Suggested revision:“LAB are valued for their probiotic properties, including modulation of gut microbiota and production of beneficial metabolites. Among them, Lactiplantibacillus plantarum is widely studied for its adaptability and health-promoting effects.”
Response: We sincerely thank the reviewer for this helpful suggestion. The redundant content in the Introduction has been condensed according to your advice. The revised text now reads: “In addition to their role in food preservation, LAB are valued for their probiotic properties, in-cluding modulation of gut microbiota and production of beneficial metabolites [2,3]. Among them, Lactiplantibacillus plantarum (formerly Lactobacillus plantarum) is widely studied for its adaptability and health-promoting effects [4,5].”
Lines 83–92, 135–144, 169–171: Instruments and brand names are repeated excessively (e.g., “NanoDrop 2000 spectrophotometer (Thermo Fisher Scientific, Wilmington, DE, USA)”). List models/manufacturers only once. Example: “DNA purity was assessed using a NanoDrop spectrophotometer (Thermo Fisher).”
Response: We thank the reviewer for this practical suggestion. All repeated mentions of instrument models and manufacturers have been streamlined. In the revised manuscript, each device is now listed only once, and subsequent references include only the model and company name. As these are well-known manufacturers, this level of detail is sufficient for clarity and does not affect reader comprehension.
Lines 183–189: Only ANOVA/Tukey is mentioned. For RNA-seq, DESeq2 usually applies multiple testing correction. Add: “For RNA-seq, differential expression analysis was conducted using DESeq2 with Benjamini–Hochberg correction for multiple testing (adjusted p < 0.05 considered significant).”
Response: We sincerely thank the reviewer for the professional and detailed suggestion. We have added the recommended description to clarify the statistical approach for RNA-seq and DESeq2 analysis: “For RNA-seq, differential expression analysis was conducted using DESeq2 with Benjamini–Hochberg correction for multiple testing (adjusted p < 0.05 considered significant).
Lines 427–440: The discussion repeats results (“protein supplementation led to higher EPS yield”) without deeper interpretation. Expand on industrial applications, e.g., how nitrogen source manipulation could improve dairy fermentation texture and EPS yields.
Response: We sincerely thank the reviewer for the constructive suggestion. Additional information has been incorporated into the Discussion to expand on the industrial relevance of our findings. The revised section now emphasizes that both plant-based and dairy fermentations may benefit from optimizing nitrogen sources, as this strategy could improve EPS productivity, texture, and overall product quality, while also supporting the development of functional fermented foods with enhanced sensory and health-promoting attributes. The added discussion is following: “From an application perspective, these results suggest that adjusting the type and level of nitrogen sources may serve as a practical means to fine-tune microbial metabolism and enhance EPS productivity in industrial fermentations. The higher EPS yield observed with soybean proteins highlights the potential of plant-based protein supplements to improve the rheological and textural properties of fermented products. Such an approach could be particularly valuable for dairy and plant-derived fermentations, where optimized EPS production contributes not only to product quality and stability but also to the development of functional foods with added health benefits.”
English clarity
Line 46: “...can facility unlocking...” → “...can facilitate unlocking...”
Response: Thank you. It is corrected.
Line 431: “EPS yield is not solely dependent on and regulated by carbohydrate nutrients...” → “EPS yield is not solely dependent on carbohydrate nutrients...”
Response: Thank you. It is corrected.
Minor Comments
Line 12: “Analyzing… via molecular and bioinformatics tools holds significant importance...”→ “Analyzing… using molecular and bioinformatics tools is important...”
Response: Thank you. It is revised as recommended.
Line 54: “candidate for functional food development”→ “a promising probiotic candidate for functional food development.”
Response: Thank you. It is revised as recommended.
Line 143: “The pH of each sample at each time point was measured...” → “The pH of samples was measured...”
Response: Thank you. It is revised as recommended.
Line 254: “demonstrated a significantly higher population compared with the other three groups.” → “demonstrated significantly higher viable counts than the other three groups.”
Response: Thank you. It is revised as recommended.
Line 407: “These adaptations underline its potential...” → “These adaptations highlight its potential...”
Response: Thank you. It is revised as recommended.
Line 37: “Lactic acid bacteria (LAB) are a diverse group...” → (LABs)
Response: We thank the reviewer for this careful observation. The abbreviation has been corrected to “(LABs)”, and all similar abbreviations throughout the manuscript have been reviewed to ensure consistent and proper singular–plural usage.
Line 39: “LAB has long been utilized in food production...” → LABs have long been utilized...
Response: Thank you. It is revised as recommended.
References: Some journal names are not abbreviated in Foods style (e.g., “Food Bioscience” → “Food Biosci.”). Please check conformity.
Response: Thank you. All journal titles in the reference list have been carefully checked and revised to conform to the journal formatting requirements.
